# Expression of pH-Sensitive GPCRs in Peritoneal Carcinomatosis of Colorectal Cancer—First Results

**DOI:** 10.3390/jcm12051803

**Published:** 2023-02-23

**Authors:** Philipp von Breitenbuch, Bernadett Kurz, Susanne Wallner, Florian Zeman, Christoph Brochhausen, Hans-Jürgen Schlitt, Stephan Schreml

**Affiliations:** 1Department of Surgery, Elblandklinikum Radebeul, 01445 Radebeul, Germany; 2Department of Dermatology, University Medical Center Regensburg, Franz-Josef-Strauß-Allee 11, 93053 Regensburg, Germany; 3Center for Clinical Studies, University Medical Center Regensburg, 93053 Regensburg, Germany; 4Institute of Pathology, University Medical Center Regensburg, 93053 Regensburg, Germany; 5Department of Surgery, University Medical Center Regensburg, Franz-Josef-Strauß-Allee 11, 93053 Regensburg, Germany

**Keywords:** colorectal, peritoneal carcinomatosis, tumor microenvironment

## Abstract

Solid tumors have an altered metabolism with a so-called inside-out pH gradient (decreased pH_e_ < increased pH_i_). This also signals back to tumor cells via proton-sensitive ion channels or G protein-coupled receptors (pH-GPCRs) to alter migration and proliferation. Nothing, however, is known about the expression of pH-GPCRs in the rare form of peritoneal carcinomatosis. Paraffin-embedded tissue samples of a series of 10 patients with peritoneal carcinomatosis of colorectal (including appendix) origin were used for immunohistochemistry to study the expression of GPR4, GPR65, GPR68, GPR132, and GPR151. GPR4 was just expressed weakly in 30% of samples and expression was significantly reduced as compared to GPR56, GPR132, and GPR151. Furthermore, GPR68 was only expressed in 60% of tumors and showed significantly reduced expression as compared to GPR65 and GPR151. This is the first study on pH-GPCRs in peritoneal carcinomatosis, which shows lower expression of GPR4 and GPR68 as compared to other pH-GPCRs in this type of cancer. It may give rise to future therapies targeting either the TME or these GPCRs directly.

## 1. Introduction

Acidosis is a common physical hallmark of solid tumors. Tumor growth and metastasis essentially require changes in the microenvironmental pH value [1,2,3,4,5]. In contrast to other cells, cancer cells seem to be better adapted to a lower extracellular pH value (pH_e_) and can take advantage of this. Nevertheless, the intracellular pH (pH_i_) of cancer cells needs to be within the physiological range, otherwise cell death occurs due to cytotoxic acid [6,7]. Normal cells show a pH_e_ of 7.2–7.4 and a pH_i_ of 6.9–7.2. Most tumor tissues have a lower pH_e_ (6.2–7.0) and higher pH_i_ (7.2–7.7). This phenomenon is called the reversed (= inside-out) pH gradient [1,2,4,5]. Cancer cells use different mechanisms to maintain this pH gradient. Compared to non-tumor cells, many cancer cells obtain their energy from a significantly increased glucose metabolism. This means that the tumor cells obtain their energy mainly from glycolysis. What is special about tumor cells is that this form of energy generation takes place not only via anaerobic but also via aerobic glycolysis (Warburg effect) [8]. The resulting lactate is removed from intracellular to extracellular space by monocarboxylate transporters (MCTs 1 + 4) [2,4]. Neri et al. report on more key pH regulators in cancer cells including isoforms of the carbonic anhydrases 2, 9, and 12 (CA2, CA9, CA12), the Na+/H+ exchanger 1 (NHE1) and the plasma membrane proton pump vacuolar ATPase (V-ATPase) [2,4].

G protein-coupled receptors (GPCRs) with their heptahelical transmembrane structure [5,9] trigger different intracellular signaling cascades after receptor activation. Only a few of the approximately 800 known GPCRs—for example, GPR4, GPR65 (TDAG8, T-cell death-associated gene 8), GPR68 (OGR1, ovarian cancer GPCR1), and GRP132 (G2A, G2 accumulation protein)—are involved in sensing pH changes through the protonation of hydrogen bonds between histidine residues. Recently, another two pH-sensitive GPCRs were found: GPR31 and GPR151 [10,11,12]. It has been shown that these GPCRs are activated via a decrease in pH_e_ through the protonation of hydrogen binding between the histidine residues. Moreover, it is thought that these GPCRs are involved in cancer cell proliferation, metastasis, angiogenesis, apoptosis, immune cell function, and inflammation [5,13,14,15,16,17,18,19,20,21,22].

Proton-sensing GPCRs are expressed in various tissues of solid tumors, but there is still a lack of knowledge about their expression in peritoneal carcinomatosis. For this reason, we investigated the expression of the proton-sensing GPCRs (GPR4, GPR65, GPR68, GPR132, and GPR151) in peritoneal tumor spots of colorectal origin.

## 2. Materials and Methods

### 2.1. Tissue Samples

All patients showed a histological proven peritoneal carcinomatosis of colorectal origin (colorectal adenocarcinoma). Paraffin-embedded tissue samples from peritoneal tumor spots were taken. For all experiments, we used tissue samples older than 10 years from the Department of Pathology at the University Medical Center Regensburg. Handling of human tumor tissue older than 10 years was approved by the ethical committee of the University of Regensburg. Under German law, the tumor tissue left after surgery after the final diagnosis can be discarded after 10 years or is free to use.

### 2.2. Immunohistochemistry

Tissue samples (embedded and fixed in paraffin) were cut into 3 µm thick sections using a microtome and then fixed on slides. Each slide was also stained with hematoxylin and eosin. This and all other subsequent staining steps were performed at room temperature.

We removed paraffin from the tissue sections by incubating them for 60 min at 72 °C, and then we rehydrated the slides with decreasing alcohol concentrations as follows: 3 × xylol for 10 min, 2 × 100% ethanol for 5 min, 2 × 96% ethanol for 5 min, 2 × 70% ethanol for 5 min. To prevent false-positive results, endogenous peroxidase was blocked with 3% H_2_O_2_ (Fisher Scientific, Waltham, MA, USA, No. 1404697) for 10 min. Simultaneously, an acidic citrate buffer with pH 6 (Zytomed, Bargteheide, Germany, REF ZUC028) was boiled for 30 min. The slides were washed in distillated water and then boiled for 20 min in the precooked citrate buffer, followed by cooling on ice for 20 min. Subsequently, they were transferred to PBS (Sigma-Aldrich, Darmstadt, Germany, No. D8537) for 10 min. Afterwards, slides were fixed with cover slides and once again washed with PBS. To avoid unspecific antibody binding, proteins were blocked with blocking solution (ZytoChem Plus HRP Kit/Rabbit, Zytomed, Bargteheide, Germany, REF HRP060-Rb) for 10 min. Next, tissue sections were incubated with polyclonal primary antibodies against GPR4 (rabbit anti-human GPR4; 1:200; Abcam, Cambridge, UK, anti-GPCR GPR4 antibody, ab188606), GPR65 (1:500; Abcam, Cambridge, UK, anti-GPCR GPR65 antibody, ab188907), GPR68 (1:50; Abcam, Cambridge, UK, anti-OGR1 antibody, ab188964), GPR 132 (1:60; Abcam, Cambridge, UK, anti-GPCR G2A antibody, ab116586), GPR151 (1:400; anti-GPCR GPR151 antibody, Life Technologies, Waltham, United States Cat.Nr.PA532803), or isotype control antibody (1:200, Abcam, Cambridge, UK, rabbit IgG polyclonal isotype control, ab27478) in antibody diluent (Zytomed Systems GmbH, Berlin, Germany) overnight at 4 °C.

The following day, the slides were washed three times with PBS. The tissue sections were then incubated with the secondary biotinylated antibody for 30 min, they were washed again three times with PBS, then incubated with streptavidin–HRP conjugate for 20 min and washed 3× with PBS. Positive controls were stained with AEC plus (Dako, Santa Clara, CA, USA, No. K 3469) until the expected staining appeared. The reaction was stopped with distillated water, and positive controls were counterstained with Mayer’s Haemalm (Roth, Karlsruhe, Germany, No. T865.3). The slides were scanned with PreciPoint M8, and the digital images were edited with ViewPoint online (PreciPoint, Freising, Bavaria, Germany).

Positive and negative controls for immmunohistochemistry of GPR4, GPR65, GPR132, and GPR151 were published recently by our group [23,24].

### 2.3. Rating

A pathologist assessed the staining of the sections visually. Sections were labeled as “++” for strong positive reactions with >80% of cells being positive and/or when staining intensity was high, “+” for 20–80% of cells demonstrating a weak positive/partial positive reaction, and “−” for <20% of cells displaying weak staining (a negative reaction). Tumors with inconsistent staining were scored as weakly positive.

### 2.4. Statistics

First, all rating results for all entities were compared using Kruskal–Wallis tests. For NCN and MMs, epidermal and dermal portions were separately used for testing. Pairwise comparisons were made via Bonferroni tests. Secondly, pairwise comparisons of BCCs vs. SCCs and of MMs vs. NCN were made for each protein using a Mann–Whitney U test, and the results are given as exact significance (shown as 2 * (1-tailed significance), not corrected for ties, for BCCs vs. SCCs and epidermal portions of NCN/MMs) or asymptotic significance (2-tailed, for dermal portions of NCN/MMs).

## 3. Results

An overview of the results can be found in Table 1. Representative immunohistochemical stainings are shown in Figure 1. All samples of each patient are depicted in Appendix A.

### 3.1. GPR4

The tissue samples from 7/10 patients showed no expression of GPR4. A weak positive expression can be seen in two patients and a strong expression pattern can only be detected in one patient.

### 3.2. GPR65

One hundred percent of our tissue samples showed a positive expression pattern for GPR65. Of these, 80% were strongly positive, 20% showed a weak positive expression.

### 3.3. GPR68

GPR68 expression could be detected in 60% of our cases, whereby in 30% the expression pattern was strong. In contrast, no expression could be detected in 40% of the histological samples.

### 3.4. GPR132

The evaluation of GPR132 expression profile showed in 70% a strong expression pattern on the surface of tumor cells. One patient showed a moderate GPR132 expression, whereas in two patients no expression of GPR132 could be found.

### 3.5. GPR151

One hundred percent of the tissue samples had a strong positive expression of GPR151.

### 3.6. Comparison of pH-GPCR Expression

Table 2 shows the statistical results. GPR4 expression was significantly lower than that for GPR65, GPR132, and also GPR151. No difference in expression was found between GPR4 and GPR68. The latter, GPR68, also showed significantly reduced expression as compared to GPR65 and GPR151.

## 4. Discussion

In our study, we investigated for the first time the expression of different GPCRs in peritoneal carcinomatosis tissue samples of colorectal origin. Our results show a strong expression for GPR151, GPR65, and GPR132. GPR68 was clearly expressed in 60% of the tissue samples, whereas GPR4 expression could only be seen in very few tissue samples. These pH-sensitive GPCRs are activated via protonation [17] and they are involved in a variety of processes, such as cancer cell proliferation, metastasis, angiogenesis, apoptosis, immune cell function, and inflammation. As a result of being influenced by such varied processes, it is clear that the effect of the different GPCRs can even be opposite to one another, for example, either promoting or suppressing tumor growth [5,13,15,16,17,18,19,20,21,22].

### 4.1. GPR4

Increased GPR4 expression is known to induce an inflammatory response in human vascular endothelial cells [5,15]. Reduced GPR4 signaling impairs the growth of murine tumor allografts [5,21]. Bai et al. found that GPR4 may function via the WNT pathway molecule transcription factor 7 (TCF7). Downregulation of GPR4 leads to a downregulation of TCF7, inhibiting cell growth and cell invasion, and promoting apoptosis of ovarian cancer cells [25]. Furthermore, TCF7 plays an important role in CRC. Long non-coding RNA TCF7 (IncTCF7) is known to be highly expressed in CRC cell lines compared to normal colonic epithelial cells, and has been shown to play a critical role in human CRC development and progression. TCF7 overexpression could promote migration and invasion in CRC cells. In contrast, TCF7 knockdown significantly inhibited migration and invasion of CRC tumor cells [26]. For the development of a peritoneal carcinomatosis, migration and invasion of tumor cells are mandatory. With respect to our results, GPR4 expression was surprisingly undetectable in nearly all samples. The reason for this remains unclear.

### 4.2. GPR65

An overexpression of GPR65 promotes glucocorticoid-induced apoptosis in mouse lymphoma cells [5,27]. Upregulated GPR65 suppresses intestinal inflammation and reduces the risk of developing colitis-associated colorectal cancer in an experimental mouse model [28]. In contrast, it has been reported that GPR65 expression enhances tumor growth in Lewis lung carcinoma cells [5,29], and that GPR65 is overexpressed in glioblastoma, which is associated with an unfavorable clinical outcome for patients [30]. In addition, Li et al. investigated the role of long non-coding RNA GPR65-1 (lincRNA) in the progression of gastric cancer [31]. They found that linc-GPR65-1 was significantly upregulated in gastric cancer tissue compared to corresponding normal tissues, and that the increased linc-GPR65-1 expression was significantly associated with a poorer TNM stage, larger tumor size, presence of distal metastasis, and poor prognosis for gastric cancer patients. Moreover, they observed that linc-GPR65-1 could regulate the PTEN-AKT-slug signaling pathway, and that this pathway might act as a tumor promotor [31]. Currently, no comparable data exist for colorectal cancer or peritoneal carcinomatosis. Due to the strong expression of GPR65 we observed in our study, one could speculate that, compared to gastric cancer, at least the PTEN-AKT-slug signaling pathway is activated by GPR65 in peritoneal carcinomatosis of CRC. This activated signaling pathway might also act as a tumor promotor in peritoneal carcinomatosis of colorectal origin.

### 4.3. GPR68

GPR68 also belongs to the known proton-sensing G protein-coupled receptors involved in pH changes during development of different tumors, e.g., neuroendocrine tumors, pheochromocytomas, cervical adenocarcinomas, endometrial cancers, medullary thyroid carcinomas, and pancreatic adenocarcinomas, whereby often tumor capillaries are strongly GPR68 positive [32]. Furthermore, GPR68 may play a crucial role in tumor biology, including tumorigenesis, tumor growth, and metastasis [33]. In human ovarian cancer cells, GPR68 improves tumor cell adhesion to extracellular matrix, but surprisingly inhibits cell proliferation and migration [5,19]. Cancer-associated fibroblasts (CAFs) are responsible for a dense fibrotic tumor stroma, and an increased expression of GPR68 could be shown in CAFs, resulting in enhanced interleukin-6 (IL-6) expression via a cAMP/PKA/cAMP response element-binding protein signaling pathway [34]. The fact that GPR68 is able to increase IL-6 expression is important since IL-6 plays a fundamental role in tumor progression [35]. Besides its expression in pancreatic CAFs [34], GPR68 is also expressed in GIST CAFs [34], appendiceal CAFs [34], and colorectal CAFs [36]. The important role of GPR68 in tumorigenesis in colorectal cancer was reported by Horman et al. First, they showed that GPR68 was one of the most highly upregulated genes when CCD-18Co (human colon fibroblast) cells were activated to CAFs after co-culture of CCD-18Co and HCT116 cells (human colorectal carcinoma cells) in 3D spheroid microtumor structures. Second, they identified GPR68 as being the number one factor in regulating microtumor formation. Finally, in their in vivo studies, MC-38 murine colorectal carcinoma cells were injected into WT and GPR68 KO mice. During the first 14 days, tumor growth was reduced in the GPR68 KO group and tumors were more fibrotic, less vascular, and had more pronounced borders. Overall, they concluded that GPR68 promotes colorectal tumor initiation in mice [36]. Based on the data of these other authors, GPR68 expression would actually have been expected in all of our patients. Surprisingly, we found GPR68 overexpression in only 6/10 patients. Therefore, one could conclude that GPR68 does not play the same role in the growth of peritoneal carcinoma tumor nodes as in the primary tumor.

### 4.4. GPR132

The anti-tumorigenic effect of GPR132 is explained by cell cycle arrest at the G2 stage [5,37] although, in contrast, GPR132 nonetheless has a strong pro-tumorigenic effect influencing tumor–macrophage communication. It was Chen et al. who showed that GPR132 mediates the reciprocal interaction between cancer cells and macrophages. Activation of GPR132 leads to an increase in lactate in the acidic tumor milieu (lactat-GPR132 axis), and activates the alternative macrophage (M2)-like phenotype, which in contrast to the “normal” macrophages facilitates cancer cell adhesion, migration, and invasion. The clinical context between GPR132 expression, M2 macrophages, metastasis, and a poor prognosis in patients with breast cancer is clear [38]. Another type of immune cell influenced by GPR132 is monocytes. With respect to the energy-transfer mechanism in the inflammatory tumor microenvironment, human THP-1 monocytes take up lactate secreted from tumor cells. High levels of lactate in turn activate hypoxia-inducible factor 1a, which promotes key enzymes of prostaglandin E2 synthesis, thereby promoting the growth of human colon cancer HCT116 cells. Interestingly, only human monocytic cells affected by lactate could stimulate the colon cancer cells. Lactate itself could not accelerate tumor growth directly [39]. In our study, GPR132 is highly expressed in all tissue patterns. This shows the important relevance of GPR132 in peritoneal carcinomatosis tumor nodules of colorectal origin. This high overexpression of GPR132 probably represents the necessary strong interaction between peritoneal carcinomatosis and cells of the immune system.

### 4.5. GPR151

Recently, it was shown that GPR151 also belongs to the one family of GPCRs that is responsible for the sensing of extracellular protons. Both GPR31 and 151 are activated under acidic conditions and their activation is maximal at approximately pH 5.8 [11]. GPR151 is involved in the nervous system of vertebrates. An upregulation of GPR151 mRNA was found in the trigeminal ganglions, causing neuropathic pain, and it was shown to bind to Gai protein, activating the extracellular signal-regulated kinase (ERK) and resulting in an ERK-dependent neuroinflammation [40]. It could, however, be proven that GPR151 couples with the G-alpha inhibitory protein to reduce cyclic adenosine monophosphate (cAMP) levels in the cells [41]. In our study, GPR151 was massively upregulated in all immunohistochemistry patterns, but its role in tumorigenesis still remains unclear. Until now, only Schreml et al. have investigated GPR151 expression in tumors. They observed highly expressed GPR151 levels in squamous cell carcinomas [23]. Our study is the first to describe high GPR151 levels in peritoneal carcinomatosis tumor tissue. Since the expression of GPR151 was very strong in all of our immunohistochemical stainings, GPR151 seems to play a mandatory role in the formation and development of a peritoneal carcinomatosis of the colorectal carcinoma.

## 5. Conclusions

pH dysregulation is a hallmark of solid tumors. The proton-sensing GPCRs are involved in sensing extracellular acidity. Cancer cells are able to use this altered environment to their advantage. We could demonstrate, for the first time, that GPCRs are extensively expressed in cancer tissue of peritoneal carcinomatosis of colorectal origin. The small number of patients is the main limitation of the study. Results of our study may help to obtain more insight into the role of GPCRs concerning peritoneal carcinomatosis. However, GPR4 and GPR68 were significantly less expressed in peritoneal carcinomatosis than other pH-GPCRs. Targeting these GPCRs and may be an attractive therapy for patients suffering from peritoneal carcinomatosis. Such a new anti-cancer therapy can probably be embedded in a multidisciplinary therapy regime. Therefore, more studies are needed to assess the specific roles of the different GPCRs in tumor biology and tumorigenesis of peritoneal carcinomatosis.

## Figures and Tables

**Figure 1 jcm-12-01803-f001:**
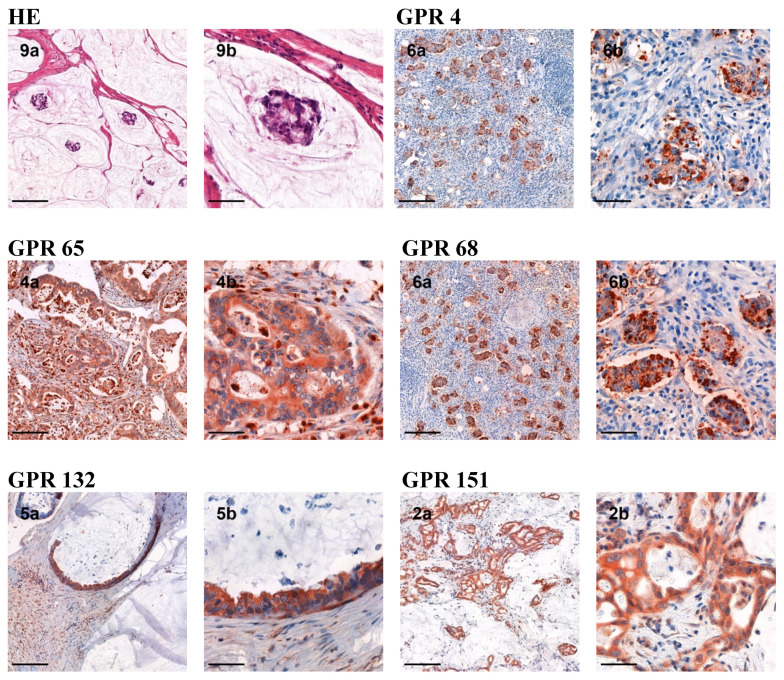
Expression of GPR4, GPR65, GPR68, GPR132, and GPR151 of peritoneal carcinomatosis of colorectal cancer. The number on each slide corresponds to the patient number (compare also Table 1). “(a)” represents a 10-fold magnification (scale bar 200 µm) and “(b)” a 40-fold magnification (scale bar 50 µm).

**Table 1 jcm-12-01803-t001:** Summary of the immunohistochemical staining results for GPR4, GPR65, GPR68, GPR132, and GPR151.

Patient	Tumor	GPR4	GPR65	GPR68	GPR132	GPR151
1	A	-	+	-	-	++
2	A	-	++	+	++	++
3	A	-	++	-	-	++
4	C	+	++	+	++	++
5	C	-	++	-	++	++
6	C	++	++	++	++	++
7	C	-	++	++	++	++
8	C	+	++	++	++	++
9	R	-	+	-	+	++
10	R	-	++	+	++	++

Primary tumor entity: A = Appendix, C = Colon; R = Rectum cancer. Immunohistochemical expression: “++” strong expression; “+” weak positive expression; “-” no expression (compare “Rating”).

**Table 2 jcm-12-01803-t002:** Statistical analysis of GPR4, GPR65, GPR68, GPR132, GPR151.

Pairs	* p * -Value	Adj. *p*-Value (Bonferroni)
GPR4 vs. GPR68	0.322	1.000
GPR4 vs. GPR132	0.011	0.109
GPR4 vs. GPR65	0.001	0.015
GPR4 vs. GPR151	<0.001	0.004
GPR68 vs. GPR132	0.120	1.000
GPR68 vs. GPR65	0.028	0.284
GPR68 vs. GPR151	0.011	0.109
GPR132 vs. GPR65	0.525	1.000
GPR132 vs. GPR151	0.322	1.000
GPR65 vs. GPR151	0.724	1.000

Comparison of all entities. Significant results are marked with green background.

## Data Availability

Available upon reasonable request.

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
