# Peer review of "Expression of pH-Sensitive GPCRs in Peritoneal Carcinomatosis of Colorectal Cancer—First Results"

_jcm, 2023, doi:10.3390/jcm12051803_

Round 1

Reviewer 1 Report

The manuscript by Philipp von Breitenbuch et al., entitled “Expression of pH-sensitive GPCRs in peritoneal carcinomatosis of colorectal cancer”, detected the expression of GPR4, GPR65, GPR68, GPR132 and GPR151 using paraffin embedded tissue samples of patients with peritoneal carcinomatosis of colorectal origin. However, the data presented here are too preliminary

Major Comments:

1. The sample size is too small (just 10 patients). Expanding the sample size may make the results

more accurate.

2. Except for IHC, qPCR and Western blotting is also necessary for the detection of pH-sensitive GPCRs.

3. The correlation between the expression of pH-sensitive GPCRs and prognosis is necessary.

4. The correlation between the expression of pH-sensitive GPCRs and clinicopathological parameters is necessary.

5. We suggest the author to further detect the expression of pH-sensitive GPCRs in the paired primary tumor.

6. As the author mentioned, GPR4 and GPR68 were significantly less expressed in peritoneal carcinomatosis than other pH-GPCRs. The authors need to further explain the reason in discussion.

Reviewer 2 Report

Dear Authors 

I believe it will be more appropriate if the grading of immunohistochemistry were done in 4 Greece axes instead of 3. The + grade category should be splitter in 2 categories 20-50% and 50-80% to give merit to the results.

thank you

Round 2

Reviewer 1 Report

The data presented here are still too preliminary. We think that it is not suitable for publication.

Author Response

see response to editor

Reviewer 2 Report

Ok

Author Response

Thank you! See reply in the pdf.
